# Modeling the Homologous Recombination Process: Methods, Successes and Challenges

**DOI:** 10.3390/ijms241914896

**Published:** 2023-10-04

**Authors:** Afra Sabei, Mara Prentiss, Chantal Prévost

**Affiliations:** 1CNRS, UPR 9080, Laboratoire de Biochimie Théorique, Université de Paris, 13 Rue Pierre et Marie Curie, F-75005 Paris, France; sabei@ibpc.fr; 2Institut de Biologie Physico-Chimique-Fondation Edmond de Rotschild, PSL Research University, F-75005 Paris, France; 3Department of Physics, Harvard University, Cambridge, MA02138, USA; prentiss@g.harvard.edu

**Keywords:** molecular modeling, molecular dynamics simulations, homologous recombination, DNA stretching, RecA, protein filament, protein-DNA interaction, multi-scale dynamics, integrative modeling

## Abstract

Homologous recombination (HR) is a fundamental process common to all species. HR aims to faithfully repair DNA double strand breaks. HR involves the formation of nucleoprotein filaments on DNA single strands (ssDNA) resected from the break. The nucleoprotein filaments search for homologous regions in the genome and promote strand exchange with the ssDNA homologous region in an unbroken copy of the genome. HR has been the object of intensive studies for decades. Because multi-scale dynamics is a fundamental aspect of this process, studying HR is highly challenging, both experimentally and using computational approaches. Nevertheless, knowledge has built up over the years and has recently progressed at an accelerated pace, borne by increasingly focused investigations using new techniques such as single molecule approaches. Linking this knowledge to the atomic structure of the nucleoprotein filament systems and the succession of unstable, transient intermediate steps that takes place during the HR process remains a challenge; modeling retains a very strong role in bridging the gap between structures that are stable enough to be observed and in exploring transition paths between these structures. However, working on ever-changing long filament systems submitted to kinetic processes is full of pitfalls. This review presents the modeling tools that are used in such studies, their possibilities and limitations, and reviews the advances in the knowledge of the HR process that have been obtained through modeling. Notably, we will emphasize how cooperative behavior in the HR nucleoprotein filament enables modeling to produce reliable information.

## 1. Introduction

In order to understand the complex mechanisms driving biological processes, modeling studies are essential. They are an effective tool for predicting the structure of conformational substates of macromolecular assemblies throughout these processes and for studying the dynamics of already-known substates. Additionally, modeling makes it possible to simulate transitions between substates, shedding light on the elusive processes causing these changes. Modeling can also assist researchers in imagining prospective outcomes. This review illustrates how modeling studies have contributed to advance our understanding of the homologous recombination process, a complex, multi-scale phenomenon involving significant DNA distortions induced by protein nucleofilaments and characterized by its dynamic and ever-evolving nature.

Homologous recombination (HR) aims to faithfully repair DNA double strand breaks that formed either by accident or were induced in the genome, for example to generate crossing-over during meiosis [1,2,3,4]. In bacteria, HR requires the formation of nucleoprotein filaments of the RecA protein bound to a DNA single strand (ssDNA) provided from the broken DNA. The ultimate goal of HR is to align that single strand to an homologous region of an undamaged copy of the genome. If the RecBCD repair pathway is followed, that region can serve as a template for a DNA polymerase synthesizing DNA regions removed by RecBCD in 3${}^{\prime }$ of the ssDNA [5,6]. The sequence is recognized through the formation of Watson-Crick interactions with the ssDNA bases, which requires exchanging the base pairing. To achieve strand exchange, the genomic DNA (dsDNA) needs to be at least transiently incorporated in the filament. The incorporated dsDNA region adopts the helical characteristics of the filament, causing stretching and unwinding. Once initiated, the strand exchange reaction propagates along the filament. The propagation is often discontinuous since some strand exchange products dissociate before reaching the 3${}^{\prime }$ end [7,8,9]. It terminates upon reaching the 3${}^{\prime }$ extremity of the ssDNA and the HR process successfully ends if it gives way to sufficient polymerization [6]. To ensure proper alignment in biologically-relevant time scales, the process is divided into several steps that can involve catastrophic reversal of the strand exchange products to the substrate components [10]. Kinetics is therefore highly present and needs to be taken into account when investigating modeling strategies.

Another important aspect of the HR mechanism is associated with ATPase activity. HR filaments catalyze ATP hydrolysis and ATP is continuously consumed in great quantities during the whole HR process [11]. ATP hydrolysis has been associated with polymorphism of the HR filament [12]; the filament plasticity is proposed to play a role in the regulation of the HR process notably by favoring dsDNA unbinding [10,13,14], but this role still needs to be fully characterized. Finally, while in vivo the HR process is tuned or controlled by several factors, such as the salt concentration (magnesium in bacteria [15,16], calcium in higher organisms [17]) or the interaction with auxiliary proteins [3]; in vitro studies showed that most HR steps can happen in the absence of auxiliary proteins, highlighting the specific role of the nucleoprotein filaments in the HR process; all HR steps take place within these evolving complexes and therefore benefit from their structural, mechanical and dynamical properties [4,18]. During HR, genomic DNA is associated, incorporated, probed, exchanged or released anywhere within long and dynamic helical assemblies. The complexity and instability of intermediate structures make it very difficult to probe these structures experimentally. Therefore molecular modeling was often called upon to fill gaps in experimental studies or investigate possible scenarios that can occur during HR.

Over the last three decades, much has been learned about the HR process, including the following: 1 the filament topology; 2. the relative arrangements of substrates within the filament; and 3. the forces driving the reaction steps. Combined with various experimental studies, modeling has played a role in building this knowledge using methods that range from topological considerations to out-of-equilibrium molecular dynamics simulations. While modeling may easily address aspects of HR that are very challenging for experiments, several characteristics of the HR complexes require developing specific approaches. For example, the filament is an assembly of interacting helices; its study therefore requires reconciling topological aspects related to such supra-assemblies [19] with atomic-level information (e.g., residue-residue contacts) [20]. More generally, each and every structural element that make up the HR process—RecA flexible loops, distortions of the DNA strands, filament polymorphism, ATP hydrolysis, disordered RecA tails—poses a unique modeling difficulty. Figure 1 gives an overview of the modeling tools that can be used to address the components of the protein-DNA assemblies involved in HR as well as the interactions between these components. Importantly, all levels of structural information, from atomic to supra-assembly, need to be considered in their entirety to obtain clues on the HR mechanism.

In this article, we review the long history of modeling the structural aspects of RecA-induced HR reaction. We discuss the validity and limitations of the modeling methods that were used to investigate this system. We report gained knowledge, success stories and pending challenges. Finally, we argue how the specificity of the HR nucleoprotein filaments as mechanically driven molecular assemblies [21,22] explains why fundamental structural information could be revealed as far as 30 years ago using low-resolution methods and why modeling can still be expected to provide keys for the comprehension of the system.

**Figure 1 ijms-24-14896-f001:**
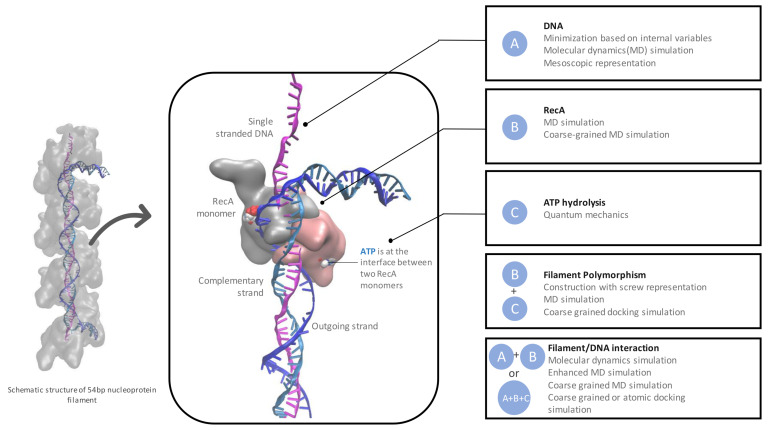
Schematic representation of the components of the homologous recombination process at different length scale levels—protein, DNA, filament, DNA-filament complex and the modeling methods that are adapted to their study. The VMD software version 1.9.4a57 [23] was used for graphical representations in all Figures in this paper.

## 2. Methods for Modeling the HR Process—Scope and Limitations

### 2.1. Molecular Modeling

Molecular modeling relies on establishing representations of macromolecules that can capture their main characteristics and be dealt with using numerical tools, based on physico-chemical principles. The representation can range from purely geometrical models and their analytical descriptors to quantum mechanics at the electronic level, via atomic-level molecular mechanics (Figure 1) (detailed principles can be found in [24]). Molecular modeling methods require a structure as a starting point, generally a conformational state of the macromolecule or the complex under study or their homologs. Integrating experimental data, be it information on residue proximity or on global properties, enables the ability to guide the conformational search towards targeted substates of the macromolecule or the complex under study. Once a substate is generated, the model gives access to additional information, such as electrostatic potential maps, interaction values and modes of deformation. If the model includes an adequate representation of the degrees of freedom and robust functional forms to describe the involved energies, it can serve as a basis for predicting possible dynamic evolution. Below is a rapid presentation of the main models and algorithms that can be used to tackle macromolecular assemblies involved in a biological process, with references to corresponding HR studies.

### 2.2. Representations, Resolution and Sampling

Each level of resolution of a macromolecule representation is associated with specific modeling methods and ranges of application. For example, quantum calculations provide access to the energies involved in chemical bond formation, breaking or polarization, along reaction pathways involving specific intermediate states [25]. The number of atoms is limited to tens or hundreds; therefore, environment effects such as solvation or ionic strength cannot be precisely accounted for, while the reaction pathways and the intermediate states need to be pre-defined. Atomic-level representations permit handling of much bigger assemblies, up to millions of atoms, but do not permit studies of chemical bond formation or breaking. Atomic-level representations are associated with the definition of force fields, an ensemble of atom type description and semi-empirical interatomic energy functions with parameters derived both from structural information and quantum mechanics calculations, which permit computation of interaction energies and forces [26,27]. This defines conformational energy landscapes that can be sampled using several classes of methods such as energy minimization, normal mode analysis, Monte Carlo or molecular dynamics simulation. The sampling variables are generally the atomic coordinates; however it can be beneficial to use internal or collective variables. If appropriately chosen, such collective variables permit to sample larger regions of the conformational space at modest computational cost and to concentrate the sampling on collective movements. This representation, combined with implicit solvent representation and the use of restraints (interatomic distance, DNA winding, …), was privileged in the nineties to develop energy minimization methods that proved very efficient to simulate cooperative transitions in DNA [28,29], at a time where computer resources (power and memory) were scarce. Another way to minimize the use of computer resources is to further decrease the resolution and use coarse-grained force fields, where the molecular unit is a pseudo-atom formed by the grouping of several heavy atoms. This also enables computational biochemists to address very large systems without inferring the choice of variables. A level of resolution where pseudo-atoms group four to five heavy atoms can capture the main steric, electrostatic and even dynamic characteristics of protein-protein or protein-nucleic acid assemblies with a good approximation when compared to atomic representation [30,31]. Coarse-grained models have been profitably associated with macromolecular docking methods [30,32,33,34] using for example multi-minimization or Monte-Carlo search as sampling strategies to explore possible assembly modes of large assemblies such as those encountered in the HR process [20,35,36].

### 2.3. Molecular Dynamics Simulations

When addressing macromolecular complexes at the atomic level, the most robust method is molecular dynamics (MD) simulation [37,38,39,40,41]. It is built on force fields that were optimized over decades (see for example [27,42]) to the point where simulations are now routinely used to fill spatial or temporal gaps in experimental information [22,26]. The principle is to compute forces on each atom resulting from its interactions with the rest of the system, to let the atoms change their position in response to the forces using Newton’s equations of motion during very short time steps (femtoseconds) and to reiterate the process. The maturity of the molecular modeling domain was acknowledged in 2013 by the attribution of the Nobel Prize in Chemistry to Martin Karplus, Michael Levitt and Ariel Warshel [43,44,45,46]. Atomic-level molecular dynamics simulations require huge amount of computer resources; their use gained in power mostly during the last fifteen years, when the spectacular rise in computer power allowed simulating medium-size macromolecules in microseconds or large macromolecular assemblies in hundredths of nanoseconds.

Molecular dynamics simulations permit the exploration of the conformational space available to a macromolecule or macromolecular complex at equilibrium, giving access to the vibrational or anharmonic movements that dissipate thermal energy. In systems where activation barriers are low, the system can visit different conformational substates during a MD trajectory. When the number of accessible substates is high, the conformational space may be too large to be fully explored during the simulation time, making absolute thermodynamic quantities such as entropy or free energy still out of reach of the calculations. Methods have been developed, though, to evaluate free energy differences, reviewed for example in [47]. In the presence of high energy barriers, enhanced molecular dynamics methods such as accelerated dynamics, replica exchange dynamics or metadynamics have been developed to help the system escape its initial substate and favor transitions towards other states [48,49]. As will be seen below, studies designed to unravel the kinetic steps in the HR process often need to address the crossing of energy barriers. Interestingly, both the high tension in the DNA strands bound to the filament, which favors cooperative movements [21], and the steric hindrance inside the HR nucleoprotein filaments [50] provide restrictions to the conformational space accessible to the system, making predictions more accessible to modeling studies. This explains how, even using low-resolution representations, important results could be simulated with a high degree of confidence that were later confirmed experimentally (see Section 3 below).

### 2.4. Mesoscopic Representation

We conclude this overview with what may be called mesoscopic representations, where the molecules under study, either DNA strands or proteins, are represented as homogeneous material submitted to physical laws of mechanics; for example the DNA being considered as an elastic rod or the protein as space filling volumes. Here, when studying the actors in the HR process, because of the extreme tension the DNA strands are submitted to and the topological and space restriction constraints, such models could provide accurate answers to complicated problems [51] and reveal surprisingly successful in predicting major structural features of the nucleoprotein filaments [52].

### 2.5. Modeling Strategies

Two comments can be made at this point. The first one concerns the choice of adequate computational methods when addressing the HR process. We have seen that each modeling method spans specific length and time scales. Since the HR process relies on the coordination of probabilistic steps occurring over large amplitude length and time scales, investigating its mechanism will require the contribution of different computational methods together with ways to navigate between the different scales. The second comment concerns the synergistic use of experiment and structural modeling. As already stated, modeling requires experimentally determined structures as starting points. When investigating macromolecular associations or allosteric phenomenons, modeling will benefit from any available experimental information, for direct use as restraints or for guiding the modeling strategy. In turn, modeling results can inspire new experiments or help set strategies for experimental studies. In that spirit, the modeling results described in the next paragraph will often refer to associated experimental results.

## 3. Contribution of Modeling Studies in Building Structural Knowledge of the HR Process

Molecular modeling has been utilized since the early times of studying the HR process, notably because structural and topological puzzling observations rapidly appeared as elements that would be essential for a complete understanding of the mechanism. The observations came from biophysical approaches, mostly electronic microscopy for the protein part and single molecule studies for DNA. The relationship between the HR components benefited from additional biochemical (e.g., cross linking studies) or biophysical information from neutron scattering, AFM, fluorescence anisotropy or fluorescence resonance energy transfer [53,54,55,56,57,58,59,60,61,62]. Information from these experiments generally could not unambiguously provide clues by themselves, but their combined use within a model could lead to important advances. We note here that after a first decade dedicated to the study of RecA-promoted HR, part of the modeling studies turned to its archaea or eukaryotic homologs, RadA, Rad51 or DMC1. We will not address these studies and will restrict this review to the bacterial HR process, where notably auxiliary proteins play a lesser role.

### 3.1. Modeling the Separate Components of the Nucleoprotein Filament Central to HR

#### 3.1.1. DNA and Its Stretched Forms

The observation that the DNA in the RecA filament is stretched by 50% and unwound by 40% with respect to physiological B-DNA stimulated the development in the nineties of single molecule micromanipulation experiments that aimed at investigating the mechanical properties of the DNA double helix, as may be conducted for any macroscopic material. These experiments provided information on DNA elastic or inelastic response to stretching, winding or strand pulling [63,64]. The results have been discussed in several stimulating reviews, which we invite the interested reader to refer to [65,66,67,68]. As an important outcome, these studies showed that double-stranded DNA responds to stretching perturbations by increasing its length up to twice its initial value. Two interpretations were proposed to account for this inelastic response: the DNA would either melt, giving way to two separate strands that lost their Watson-Crick binding, or collectively undergo a structural transition towards what was called an S-form (Figure 2A). As shown by Albrecht et al. [69], both pathways can be followed depending on the applied stretching load.

Minimization methods associated with DNA representation based on internal and helical variables appeared to accurately predict the collective deformations associated with stretching-induced structural transitions [29,70]. Using such a method, the group of Lavery predicted two different outcomes when pulling the strands either from the 3${}^{\prime }$ or from the 5${}^{\prime }$ extremities [63,70] (Figure 2A, center), that were later confirmed experimentally [71]. Interestingly, this group showed that pulling the 3${}^{\prime }$ extremities unwinds the double helix; when reaching 50% extension, the DNA unwinding reaches exactly the 40% value found in the RecA filaments [72]. In addition, at 50% extension the base pairs become perpendicular to the DNA axis, in agreement to the experimental results of Nordén and Takahashi [73,74], while beyond and above the 50% value, the bases tilt in order to conserve part of their stacking interaction; switching to a perpendicular orientation permits to optimally regain the lost stacking interaction within groups of stacked bases, at the expense of introducing large intercalation sites in the filament every three base pairs [75]. Indeed, the structure resulting from their simulation in the late 1990s is strikingly similar to the structure of DNA bound to the filament, solved by crystallography ten years later [72,75,76]. That structure corresponds to the least frustrated DNA conformation which obeys the HR filament helical characteristics (Figure 2A). In addition to the minor groove widening that permits easy access to the DNA bases, that stretched form confers to the DNA unexpected properties. Recent electrostatic analysis argued that repulsion between the strand backbones is much smaller than in B-DNA [77].

Other studies explored the mechanics associated with DNA stretching using MD simulations [78,79,80,81]. Zacharias showed that applying minor groove widening restraints on a DNA double helix produced exactly the same transition in the DNA double helix as obtained when pulling the 3${}^{\prime }$ extremities [82]. A noticeable early MD exploration by McCammon on DNA deformations when embedded in a low dielectric medium, a very rough representation of the dielectric properties of the filament, also showed strong tendencies towards minor groove opening [83]. Depending on the simulation settings, MD simulation also provided access to the melting scenario and pointed to the importance of entropy (see for example [79]). Finally, Lee et al. [84] used MD simulations in coarse-grain representation to explore the importance of pre-stretching one DNA strand through its binding to a recombination filament when annealing a second DNA strand; separation of the first DNA strand into groups of three base pairs enable triplet based-annealing process to take place, thus increasing recognition efficiency [85,86]. We note however that recognition efficiency is even higher if the assembly of the two stretched strands results from RecA-catalyzed homologous recombination [77] (see Section 3.2).

**Figure 2 ijms-24-14896-f002:**
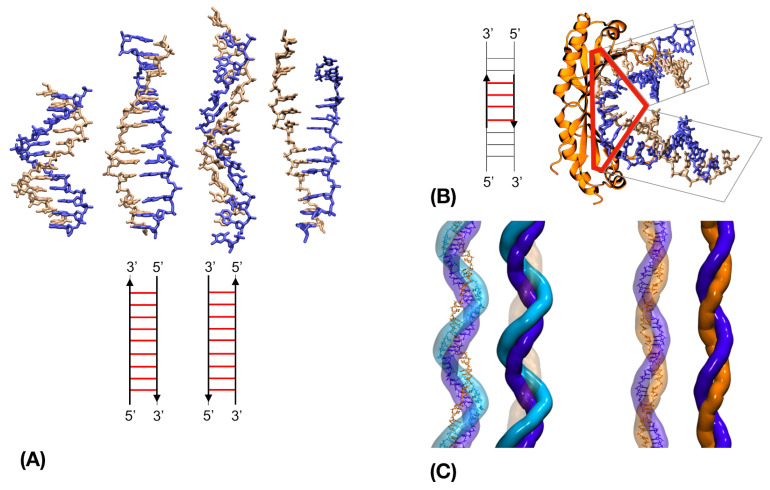
Modeling stretched DNA. (**A**) top, from left to right: B-DNA, S-form resulting from 3${}^{\prime }$-3${}^{\prime }$ pulling; S-form resulting from 5${}^{\prime }$-5${}^{\prime }$ pulling; melted DNA; bottom: schemes indicate the direction (arrows) and region (red lines) of application of the pulling force associated to the corresponding structure. (**B**) Binding to the TBP protein (orange, right panel) locally stretches the TATA-box DNA (red lines in the left panel scheme) generating kinks at the junction between relaxed (grey boxes) and locally stretched (red box) DNA regions (from crystal structure 1YTB). The DNA strands in (**A**) and (**B**) are colored blue and yellow. (**C**) Positions of the DNA strands in the synaptic (**left**) and post-synaptic (**right**) recombination filaments, after [87]; the outgoing, complementary and ssDNA strands are represented in cyan, blue and orange, respectively. The outgoing strand is not represented in the post-synaptic structure. The position of the outgoing and complementary strands in the synaptic structure correspond to axis separation values of 15 and 10 Å, respectively, as proposed by Egelman and Yu in 1989 (Figure 4 of [52], see also [88]). The positions of the heteroduplex strands in the post-synaptic structure, separated by 6 Å from the axis, were also predicted in that study.

Another important conclusion from the early modeling studies is that the structural deformation resulting from 3${}^{\prime }$3${}^{\prime }$ pulling, which involves unwinding and large minor groove widening, belongs to the same structural family as deformations locally induced in the DNA double helix by architectural proteins such as SU, SRY or TBP [89] (Figure 2B). In these complexes, the junction between the local deformations and the relaxed regions of the DNA induces a change in the DNA direction. Based on this observation, it was speculated that the entry of the dsDNA in the filament would produce a kink in the double helix [75]. The presence of such kink at the junction between the non-specifically bound and the stretched regions of the dsDNA was later confirmed experimentally by atomic force microscopy of filaments of Rad51, the eukaryotic RecA analog, with a kink angle close to 90° for the DNA entry in 5${}^{\prime }$ and a lesser value for its output in 3${}^{\prime }$ [59] (see also [90], where these kinks are referred to as 3${}^{\prime }$-tilt and 5${}^{\prime }$-tilt, respectively). As will be seen below, modeling studies identified the kink in 5${}^{\prime }$ of the RecA filament incorporated DNA as a key factor to enabling pairing exchange [87].

Finally, several studies explored possible structures for three-stranded DNA assemblies that have been proposed to be intermediate species of the HR reaction [91]. During strand exchange, three DNA strands coexist in the filament in stretched conformations. The possible involvement of these three strands in homology search and strand exchange, in the form of RecA-bound stretched triple-helices, was postulated [92] and explored via modeling studies [93,94,95]. The structural knowledge that has been established over the following two decades (for example [76,87,90,96]) excludes the formation of such triple-helices within the RecA filament, where RecA-bound DNA strands are separated by bulky L2 loops. We note however that transiently formed individual triplets of the R-form where identified during MD simulations of post-synaptic filament forms [50]. Outside the RecA filament, intertwined triple stranded structures [97] may appear during the course of the HR process as a result of ATP-driven RecA-DNA dissociation.

#### 3.1.2. RecA

RecA is a 38 kDa protein composed by three domains, an ATP-binding domain (37–270) that exhibits two flexible loops L1 (156–165) and L2 (194–212), a globular C-terminal domain (271–333) ending with a 19 amino-acid (334–352) disordered tail, and a N-terminal domain composed of a terminal helix (1–26) and a 13 amino-acid long (27–36) linker (Figure 3A). The dynamic properties of the RecA protein part of the nucleoprotein filament have been addressed in several studies, with assemblies varying from two to 17 monomers. Molecular dynamics simulations confirmed the speculation based on electron microscopy (EM) observations for the C-terminal domain and crystal structures of DNA-free filaments for the loops, namely that the C- and N-terminal domains and the L1 and L2 loops are mobile with respect to the central ATP-binding domain, which itself is quasi-rigid. To a lesser extent, flexibility was also found in the terminal loop of the LexA-binding hairpin (226–245). (Figure 3A) [50,87,98,99,100,101]. In the filament, the N-terminal helix is bound to the adjacent monomer so its movements, accompanied by the reorganization of its long flexible linker akin to a welcoming handshake, reflect the inter-monomer movements [102] (Figure 3B).

At the other extremity of the protein chain, the disordered C-terminal tail, represented in white to blue shadings in Figure 3A, was the object of specific MD simulations aimed at understanding how the tail mediates the observed influence of magnesium concentration on the rate of dsDNA uptake [103] and strand exchange [15]. The tail was found to remain mostly disordered and to transiently interact with the RecA ATP-binding core, C-terminal or N-terminal regions [15,104]. In the filament, the C-terminal tails appeared to participate in inter-subunit interactions by simultaneously binding regions from adjacent monomers. Interactions with the C-terminal domain transiently overlapped the non-specific DNA binding region where the dsDNA initially enters the filament [87,105], which evokes possible competition for binding that region. Observed interactions between the C-terminal tails and N-terminal linkers [15], or occurrence of transient tail insertion in the secondary DNA binding site inside the filament [104], may be key elements of the tail interference with the HR process.

#### 3.1.3. Polymorphism of RecA Assembly

RecA proteins self-assemble mostly as filaments. They can also form dimers, rings or bundles [106,107,108,109], but the form that is active for DNA sequence recognition and strand exchange is a right-handed helical filament bound to DNA in the presence of ATP (Figure 3B, top left). Using coarse-grained simulations with interaction energies focusing on electrostatic, stacking and repulsive interactions, Mukherjee et al. [36] explored the mechanism of ssDNA-RecA filament formation and identified RecA dimers as the elementary binding unit, as well as important roles for L2 loop flexibility and the presence of ATP to guide the ssDNA association in the first DNA binding site, site I. The HR filament became an object of pronounced interest when analyses of EM micrographs revealed its ability to exist in two main forms, the so-called “stretched” or “compressed” forms, depending on whether the cofactor is ATP or its hydrolyzed form ADP [110,111] (Figure 3B, top panel). The two forms, both right-handed helices, differ via their average pitch values, where the stretched form is 20% higher. This seemingly small difference reflects a completely different arrangement of the RecA subunits in the filament (Figure 3B, bottom). Notably, the two binding geometries present completely different sets of residue-residue contacts across the interface [20], with the exception of the contacts between the N-terminal helix and its binding interface on the neighboring monomer, which remain remarkably conserved in both forms [76,112]. The consequences of the switch in binding geometries are significant in terms of accessibility of the amino acids, electrostatic properties or topology of the filament deep groove [20]. The topology of the groove, including its width, the presence of longitudinal tracks and more generally the partitioning of the 3D space, rules the relative positions of the DNA strands [50] but also the access of partner proteins to the filament interior; for example, the LexA repressor protein only binds the stretched form of RecA filament which catalyzes its self-cleavage [113], while RecX regulates the HR process by specifically binding to the compressed form [13,114].

The filament polymorphism was addressed through different computational approaches. Wang and collaborators focused on the conformational changes of the RecA N-terminal linker across different filament forms of RecA. These forms include the stretched and compressed forms already mentioned, but also alternative forms solved by crystallography in non-physiological conditions, including an unwound and left-handed helix of a RecA homolog [115,116]. In all those forms, the N-terminal helix maintains the same interaction with the adjacent monomer, while the remaining interface between ATP-binding domains undergoes large variations [20]. The N-terminal linker adapts its conformation to changes in the relative positions of the N-terminal helix and the ATP-binding domain. Several groups investigated the possible role of the N-terminal linker in inducing the transition between stretched/compressed filaments [115,117,118]. Alternatively, Boyer et al. [20] concentrated on the interface between quasi-rigid regions of the ATP-binding domains. Using coarse-grained docking simulations, they showed that information on binding preferences is encrypted on the protein core surface; interestingly, the binding geometries corresponding to the stretched and compressed filament forms were found within the three assembly geometries identified as the most favorable for RecA/RecA binding. Other favorable interfaces correspond to a left-handed helix and various cyclic N-mers, with the number N of subunits ranging from 2 to 11 (see supplementary information in [20]). This study also defined families of oligomeric forms based on interface similarity and proposed that interface similarity may be involved in the easy switch from helical to ring geometry observed in some proteins, e.g., Dmc1, an eukaryotic analog of RecA [20,119].

#### 3.1.4. ATP Hydrolysis in HR Nucleoprotein Filaments

Polymorphism in the RecA filaments is directly related to ATP hydrolysis. Similarly to many motor assemblies that contain RecA-like ATP-binding domains, ATP cofactors are situated at the interface between consecutive monomers. ATP hydrolysis can result in interface shift towards the geometry found in the “compressed” form, where ADP is accessible and can be displaced by a new ATP molecule to feed ATP hydrolysis cycles (Figure 3B, bottom panel). The mechanism of the phosphodiester cleavage reaction (ATP hydrolysis) involves residues from both interacting monomers [120]. Reymer et al. [25] studied this mechanism at the quantum mechanics level. They characterized how networks of hydrogen bond interactions at the interface that connect ATP to the DNA binding sites are modified during ATP hydrolysis, thus lifting a veil on the question of mechanochemical energy conversion coupled to ATP hydrolysis. Boyer et al. [50] explored the possible structural implications of ATP hydrolysis at the level of the filament. They examined the hypothesis of an ATP- to ADP-interface shift in the central monomer of a two-turn filament. The interface shift produces a 30° kink in the filament axis, with modifications in the topology of the deep groove that extend over one filament turn in 3${}^{\prime }$ of the kink. Using specific construction tools [121], this study showed that the filament distortion that would result from the central interface shift are compatible with the presence of up to three bound DNA strands. In addition, a change in pairing strand partners spontaneously occurred over up to three consecutive base pairs during molecular dynamics simulations of the resulting models, thus pointing to a possible way ATP hydrolysis may induce strand exchange product destabilization [10].

### 3.2. Modeling Intermediate Species along the HR Steps

As already mentioned, HR is a dynamic process that continuously evolves in time but also in space. RecA proteins self-assemble on a ssDNA, then a dsDNA binds non-specifically anywhere on the filament and the sequence is probed; depending on the probing issue, the dsDNA can unbind and rebind elsewhere in the filament or the DNA strands can be exchanged; strand exchange progresses along the filament but can revert. All this makes intermediate states very difficult to capture.

#### 3.2.1. Modeling the DNA Strand Positions inside the HR Filament

The position of the DNA strands inside the filament has been the object of intensive research by experimental methods such as spectroscopy or cross-linking experiments, see [75]. Of note is the early proposition by the Egelman group of the relative arrangement of the three incorporated DNA strands, before and after strand exchange (corresponding to the synaptic and post-synaptic states, respectively) [52] (see Figure 2C). The models were proposed based on low-resolution EM difference maps obtained with or without bound DNA, but also using geometric and mechanical knowledge of DNA properties such as the highest possible values for backbone extension, combined with the helical characteristics imposed by the filament. The predicted strand positions were later confirmed experimentally based on fluorescence resonance energy transfer measurements, crystallography and Cryo-EM [58,76,90]. Using a combination of docking simulations and molecular dynamics simulations Yang et al. [87] produced a model of the synaptic filament at the atomic level. Remarkably, when submitting this model to out-of-equilibrium accelerated (enhanced) molecular dynamics simulation, base pairing exchange spontaneously occurred and the dsDNA resulting from strand exchange (heteroduplex) assumed the known structure of the heteroduplex [76], providing key information on the driving force and the dynamic details of the transition (described below). The strand arrangement in both the synaptic and the post-synaptic atomic models perfectly converged with the early propositions. It was further confirmed recently with the resolution of Cryo-EM structures of artificially stabilized early recombination intermediates [90].

#### 3.2.2. Modeling the Early Stage of dsDNA Incorporation in the HR Filament

In addition to the position of the strands inside the filament, other noticeable elements of the structure have been the subject of modeling studies. As mentioned earlier, the presence of kinks at the junction between the dsDNA region inserted in the filament and the non-specifically bound dsDNA region had been anticipated based on similarities of the RecA-bound DNA structure with DNA bound to architectural proteins [75]. The introduction of kinks in the dsDNA structure was the basis of a first exploratory modeling work on the early interactions between the probed dsDNA and the ssDNA-RecA filament [35]. Based on a combination of coarse-grained docking and interactive simulation, the study aimed at probing a possible association scenario where the dsDNA would directly probe the ssDNA sequence in site I (the ssDNA-binding site) before its strands would separate. Although the study concluded that the presence of kinks at either side of a dsDNA base pair triplet stretch sterically enables that stretch to align with the axis and to probe a ssDNA triplet, a molecular dynamics simulation by M. Zacharias showed no base pairing exchange in the modeled conformation (personal communication). Although this is only a hint, accounting for additional information from the physics or biology fields (notably the high search rate, which would not be compatible with displacing a L2 loop or with binding upon searching, see [86,87]) led to abandonment of the hypothesis that initial recognition occurs in the ssDNA-binding site. Exploration of the alternative pathway, where the dsDNA first binds in the secondary binding site, is described in the next paragraph. A kink at the junction between the relaxed dsDNA region in B-form and the stretched region inserted in the filament is still a key element of that model [87].

#### 3.2.3. Modeling RecA-Induced Pairing Exchange

Five years before the Cryo-EM structure of HR intermediates was published [90], Yang et al. proposed a model for the early association complex between dsDNA and the HR filament (synaptic complex) and simulated base pairing exchange within the proposed structure [87]. The model, built via a combination of atomic and coarse grained simulations, was divided into three main regions (Figure 4A): a threshold region where the dsDNA in B-form non-specifically binds the filament deep groove via strong interactions with RecA C-terminal domains as proposed by Kurumizaka et al. [105]; a region where the stretched and unwound dsDNA non-specifically binds the filament site II, in an orientation parallel to the axis; kinked regions at the junction between the relaxed and the stretched DNA regions. The model pointed at the role of the aromatic Phe203 residue of loop L2 to stabilize the kinks in the dsDNA structure, a role that corresponds to experimental observations [90,122].

Since the strongly charged site II track, separated from site I by the bulky L2 loops, is laterally displaced from the filament axis with respect to site I, the dsDNA backbone in site II is even more stretched than the ssDNA backbone in site I. A series of MD simulations showed that as the number of incorporated dsDNA bases increased, the building stress could be released either by dsDNA unbinding or by the complementary strand exchanging its pairing partner for the ssDNA. Indeed, when submitted to accelerated MD simulation, the model structure spontaneously gave way to base pairing exchange via a combination of rotation of complementary strand bases and local backbone displacement. We note that the spontaneous transition of the complementary strand backbone towards its known post-strand exchange position [76], starting from a modeled position, is another manifestation of stress-induced mechanical coupling within the DNA-RecA system. Its observation contributes to validate the modeling process.

The analysis of the system evolution during the MD simulation offered structural interpretations that explain several fundamental features of the HR process: the rapidity of the sequence check, due to the complementary strand conserving interactions with its initial partner strand during the search; initial probing limited to eight base pairs; instability of non-homologous dsDNA insertion in site II when its length exceeds 12 base pairs, unless strand exchange takes place; initiation of the search directly in 3${}^{\prime }$ of the dsDNA kink, where loosening of the complementary strand backbone enables its displacement to site I. The simulations also resulted in the stabilization of the kink in 3${}^{\prime }$ of the exchanged base pairs in an orientation strikingly similar to that found in DNA in complex with polymerases (PDB ID 4IRC [123]). This similarity served as a basis to model the association of DinB DNA polymerase with the RecA filament, described below [22,124].

The Cryo-EM structures of non-homologous or homologous association complexes [90] confirmed the most part of these numerical observations and provided details about stabilized intermediate states before and after strand exchange, although the unstable complementary strand could not be solved in the synaptic complex. Short-lived intermediate structures may be too unstable to image using Cryo-EM. Two factors stabilized structures enough to allow imaging: 1. a heterologous bubble was introduced in the outgoing strand of the post-synaptic structure (Figure 4B) to inhibit strand exchange reversal that frequently occurs in short strand exchange products [10]; 2. the filament samples were engineered to restrain their sizes to less than two filament turns. Unfortunately, those two stabilization factors may introduce biases.

One bias could arise from the introduction of a heterologous bubble in the post-synaptic structure. The bubble led to an intermediate structure resembling a post-synaptic complex (PDB ID 7JY9); however, because the dsDNA contains a heterologous bubble in the sequence region corresponding to the heteroduplex, the captured intermediate results from annealing of two unpaired ssDNA, not strand exchange. In contrast, during strand exchange, the dsDNA must locally melt before invading strand bases can Watson-Crick pair with complementary strand bases in the dsDNA; experiments suggest strand exchange and annealing follow different pathways [77]. Finally, in vivo reverse strand exchange can restore the original base pairing in the dsDNA, whereas with a heterologous bubble, reverse strand exchange cannot occur. Thus, caution is advisable when drawing conclusions about the process of strand exchange and its directionality based on this annealing product, as proposed in [90].

The short filament length may also have introduced a bias. In all solved Cryo-EM structures, the 3${}^{\prime }$ extremity of a bound dsDNA was shown to interact with the same C-terminal domain at the 3${}^{\prime }$ extremity of the mini-filament system. Curiously, this was interpreted as the dsDNA incorporation systematically starting from that exact region and progressing from 3${}^{\prime }$ to 5${}^{\prime }$. An alternative interpretation of the Cryo-EM data is that the search intermediates in the Cryo-EM structures result from the 5${}^{\prime }$-3${}^{\prime }$ progression of dsDNA incorporation reaching the 3${}^{\prime }$ extremity of the mini-filament (or being arrested by already present DNA in the case of multi-incorporations). The alternative interpretation is supported by the orientation of the kinks at the 5${}^{\prime }$- and 3${}^{\prime }$-junctions of the filament-bound heteroduplex, which match both the simulated [87] and experimentally observed [59] orientations (90° in 5’ and less than 60° in 3${}^{\prime }$, respectively). We note that the 3${}^{\prime }$ terminus of the filament is the final target of the HR process, where the progression of DNA strand exchange terminates and gives way to DNA synthesis [2,6,125]. The extra stabilization of the dsDNA binding at the 3${}^{\prime }$-terminus of the mini-filaments suggested by the Cryo-EM observations may be related to the presence of strong electrostatic potential at both extremities of the filament (Figure 5). Stabilization of dsDNA binding to the filament 3${}^{\prime }$-extremity may prove important for the relay transition between the HR process and the subsequent DNA synthesis necessary to complete DNA repair [6,124].

The complexity of the HR process and the fact it spans several orders of magnitude in terms of time and space make its unraveling prohibitive when using a unique technique, be it experimental or theoretical. Structural snapshots by themselves are not sufficient to inform about the details of dynamic steps. Their interpretation needs to integrate a wide range of published observations, including modeling studies [22,127,128]. Cryo-EM and modeling approaches, both prone to biases (the modeling possible biases were described in part 1, the Cryo-EM potential biases reside in the need to stabilize dynamic structure intermediates), provide complementary information that is beneficial to integrate.

### 3.3. Interaction with Partner Proteins

The HR filament is not only a scaffold for probing the complementary strand sequence of genomic DNA—it is also active in binding proteins that can participate in the control of the HR reaction as auxiliary factors, such as for example DinI or RecX [103,129], or be actors in the DSB repair process, such as LexA repressor or DNA polymerases. DinI stabilizes the RecA filament in a stretched form while RecX binds the compressed filament and inhibits ATP hydrolysis, LexA repressor cleavage and DNA strand exchange. LexA repressor self-cleaves when bound to the stretched form of RecA filaments, launching the transcription of genes of DNA-repair enzymes it otherwise represses. Error-prone DNA polymerases such as DinB are activated by their binding to the RecA filament [124]. Most of these proteins bind in the filament deep groove, hence the importance of the filament topology and plasticity. For example, RecX specifically binds filaments in the compressed form while LexA or DinI bind filaments in the stretched form [13,114]. Proteins that participate to the control of the HR reaction often bind filament via unstructured regions, making the construction of models a challenging task.

Models have been proposed for the Swi5-Sfr1 complex regulating the HR process in yeast by integrating small angle X-ray scattering (SAXS) data [130]. Proteins such as LexA, RecX or polymerases are stably folded. Modeling their interaction with the filament, for example using docking simulations, is therefore less of a challenge. A proposed model of the RecX protein in interaction with the RecA filament, compatible with small angle neutron scattering data, was found stable during molecular dynamics simulation [131]; however the filament was taken in its active form in that study. Before a CryoEM structure of RecA filaments decorated by LexA proteins could be solved [132], several atomic models were proposed for the interaction between LexA and the filament following characterization of the RecA/LexA interface [133,134]. Notably, an integrative modeling study using the HADDOCK docking web server [135] with cross-linking data as driving information [136] proposed an interpretation on how the filament scaffold and the accessible residues promote the protein self-cleavage.

The DinB DNA polymerase, of the Pol IV family, has been shown to bind the RecA filament 3${}^{\prime }$ terminus in order to initiate the synthesis of missing regions of the ssDNA, using the captured dsDNA complementary strand as a template. The polymerase shows low processivity, but its binding to the RecA filament enhances its activity [124] in a way that was shown to participate in the HR proof-reading process [6]. A model of DinB bound to both the RecA units and the three DNA strands resulting from strand exchange was proposed based on the crystal structure of the free DinB-DNA complex, and a model of post-synaptic nucleofilament resulting from the Yang et al. simulation [87]. Coarse grained interactive simulation was used in order to accommodate the very crowded environment of the filament deep groove to the polymerase, while keeping the polymerase structure and its interactions with the DNA strands mainly unchanged [22,124]. The resulting model shows a flexible L2 loop deeply buried in the catalytic cleft of the polymerase [22], providing a structural interpretation to puzzling previous experimental observations on DinB association with isolated RecA proteins that had identified patches of catalytic cleft residues among the strongly interacting regions [5]. Challenging the stability of the model via MD simulations revealed another manifestation of tension-induced collective rearrangements in the DinB-RecA nucleofilament system [22].

## 4. Conclusions

Because the viability of the cell is at stake when double-strand breaks occur in the genome, evolution has converged towards a universal, highly complex biological process aiming at faithfully repairing the DNA in a time that is compatible with the cell lifetime. As we learn more about the different steps of the HR process, a very dynamic picture comes to shape that involves a succession of strand exchange product formation and dissociation, strand exchange progression, filament association and dissociation. Remarkably in bacteria, all those steps necessitate only a reduced number of partner molecules: the DNA strands, RecA proteins and ATP. The main ingredients of the dynamic evolution of the HR assemblies therefore reside in the properties of the HR filament, its mechanical response to the incorporated DNA under stretching stress or its response to ATP hydrolysis. Fruitful cooperation between experiments and modeling will be needed to tackle the last enigmas linked to the HR process and access a complete understanding of the whole process.

## Figures and Tables

**Figure 3 ijms-24-14896-f003:**
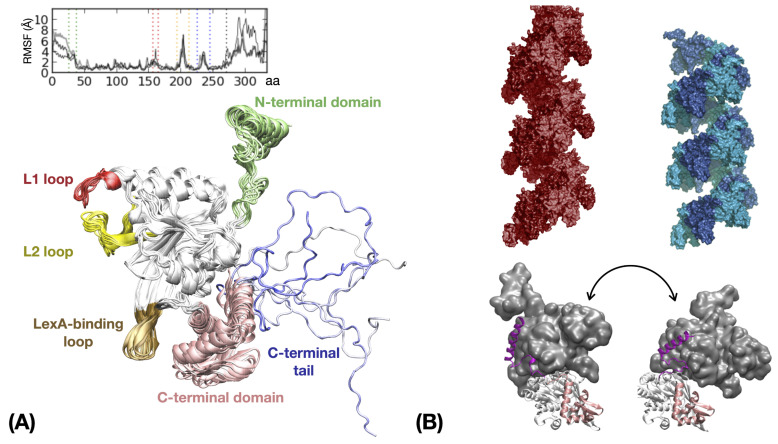
RecA protein within the recombination filament. (**A**) Dynamic behavior of a RecA protein embedded in a RecA-ATP filament during a 200 ns molecular dynamic simulation, represented by the superposition of snapshots taken every 25 ns; regions that show higher flexibility (see typical RMSF plot in insert limited to the 1–333 region, adapted from [50]) are color-coded according to the labels and the dotted lines in the insert. (**B**) RecA-RecA interface change upon ATP hydrolysis. top panel: two filament forms observed in the presence of non-hydrolyzable ATP analogs (red, **left**), ADP or no cofactor (blue, **right**); monomers are colored alternatively with light and dark shades; bottom panel: two interacting monomers are represented in each view, with the bottom monomer in cartoon (N-terminal domain in purple, C-terminal domain in pink, ATPase domain in white) and the upper monomer in surface representation. The binding geometry in the left panel corresponds to the ATP-bound geometry (ATP is embedded in the interface) while the right panel shows the binding geometry corresponding to ADP as a cofactor or no cofactor.

**Figure 4 ijms-24-14896-f004:**
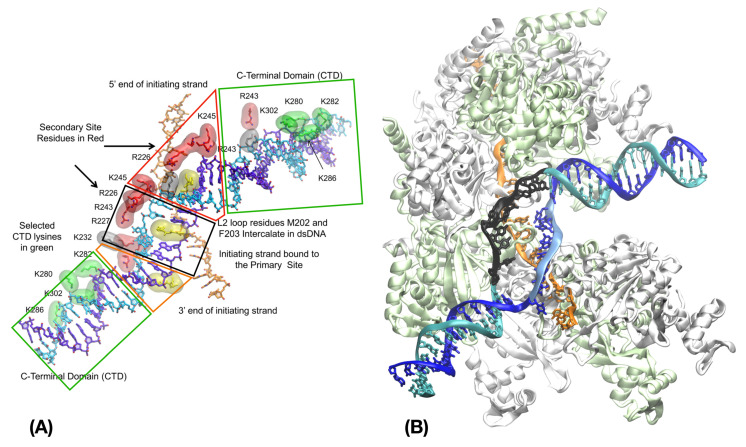
RecA filament and strand exchange. (**A**) Model of the synaptic filament, adapted from [87]; the regions corresponding to the threshold regions, the 5${}^{\prime }$-and 3${}^{\prime }$ kink and the incorporated dsDNA in site II are lined with green, red, orange and black boxes, respectively. The protein monomers are not represented except for DNA-binding residues. the dsDNA strands in the homologous region of (A) and (B) are represented in cyan and blue for the outgoing and complementary strands, respectively, while the ssDNA is in orange. (**B**) Cryo-EM Structure of the post-synaptic filament, PDB ID 7JY9 [90]; RecA monomers are shown in cartoon representation and are alternatively colored lime and white; in the dsDNA heterologous bubble region, the outgoing strand is colored in black and the complementary strand backbone is in light blue; the complementary strand bases in that region form Watson-Crick interactions with the corresponding ssDNA bases in orange.

**Figure 5 ijms-24-14896-f005:**
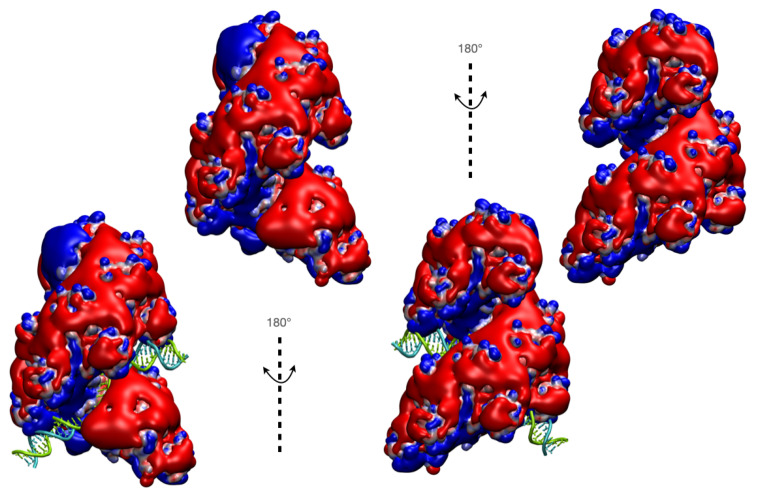
Electrostatic interactions created by RecA-ssDNA mini-filaments. (**top**) two views, rotated by 180°, of the electrostatic potential in the vicinity of the ssDNA-RecA Cryo-EM mini-filaments PDB ID 7JY9 in the absence of the dsDNA strands: surfaces corresponding to isopotential values of −0.8 and +0.8 are colored in red and blue, respectively. Strong electropositive potential observed at both ends of the mini-filament may favor DNA binding in those regions. (**bottom**) same view with the bound dsDNA shown in tube (backbone and licorice bases, with the outgoing strand in blue and the complementary strand in green). The potential was calculated using the APBS web server (https://server.poissonboltzmann.org, accessed on 10 August 2023) [126] and was represented with VMD [23].

## Data Availability

No new data was produced for this review article.

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
