# Peer review of "Modeling the Homologous Recombination Process: Methods, Successes and Challenges"

_ijms, 2023, doi:10.3390/ijms241914896_

Round 1

Reviewer 1 Report

The article submitted for review is of high interest. It is adequately written and its conclusions are clear and concise. I do not suggest any modifications to it as it seems to me to be a very well prepared article. 

Author Response

We thank the reviewer for his or her interest and nice comments.

Reviewer 2 Report

The manuscript ijms-2620827 entitled Modeling the homologous recombination process: methods, successes and challenges by Afra Sabei and coworkers, is a review article about the Homologous recombination (HR). HR involves the formation of nucleoprotein filaments on DNA single strands resected from the break. The nucleoprotein filaments search for homologous regions in the genome, and promote strand exchange between the ssDNA homologous region in an unbroken copy of the genome.

Modeling can also assist researchers in imagining prospective outcomes. This review illustrates how modeling studies have contributed to advance the understanding of the homologous recombination process, a complex, multi-scale phenomenon involving significant DNA distortions induced by protein nucleofilaments and characterized by its dynamic and ever-evolving nature.

The manuscript review deeply the available literature in the field.

The text is fluent and well written, English language is of high quality.

Figures are informative and clear.

The overall merit is excellent.

Minor revision: a linguistic revision is recommended.

The english language is of good quality

Author Response

We thank the reviewer for his or her very nice comments. We have revised the manuscript and corrected the errors or typos.

Reviewer 3 Report

The manuscript entitled ‘Modeling the homologous recombination process: methods, successes and challenges’ written by Afra Sabei et al. gathers data regarding various aspects of research focused on the homologous recombination process. The authors presented various approaches to explain the mechanisms controlling this process, together with their limitations and successes.

The manuscript is well structured, relevant to the field, and has appropriate references and informative figures. The English language is clear. I have no specific comments for this manuscript, and I recommend the editors to accept it for publication.

Author Response

We thank the reviewer for his or her very nice comments.

Reviewer 4 Report

In this article, Sabei et al. reviewed the current knowledge of the prokaryotic homologous recombination using different types of modeling strategies. Given the biological importance of the homologous recombination process and the many challenges the researches encountered, this review article would be a significant contribution to the field. This manuscript is well-organized and comprehensively described. The model figures are also clear and easy to understand. This manuscript is acceptable in its present form for publication.

English language is fine. However, thorough proofreading will be needed to correct minor typos in this manuscript. 

Author Response

We thank the reviewer for his or her nice comments. We have corrected the typos.